# Early Parental Death and Risk of Psychosis in Offspring: A Six-Country Case-Control Study

**DOI:** 10.3390/jcm8071081

**Published:** 2019-07-23

**Authors:** Supriya Misra, Bizu Gelaye, Karestan C. Koenen, David R. Williams, Christina P.C. Borba, Diego Quattrone, Marta Di Forti, Caterina La Cascia, Daniele La Barbera, Ilaria Tarricone, Domenico Berardi, Andrei Szöke, Celso Arango, Andrea Tortelli, Lieuwe de Haan, Eva Velthorst, Julio Bobes, Miguel Bernardo, Julio Sanjuán, Jose Luis Santos, Manuel Arrojo, Cristina Marta Del-Ben, Paulo Rossi Menezes, Jean-Paul Selten, Peter B. Jones, James B. Kirkbride, Bart P.F. Rutten, Jim van Os, Robin M. Murray, Charlotte Gayer-Anderson, Craig Morgan

**Affiliations:** 1Department of Social and Behavioral Sciences, Harvard T.H. Chan School of Public Health, Boston, MA 02115, USA; 2Department of Epidemiology, Harvard T.H. Chan School of Public Health, Boston, MA 02115, USA; 3Department of Psychiatry, Boston Medical Center, Boston University School of Medicine, Boston, MA 02118, USA; 4Social, Genetic and Developmental Psychiatry Centre, Institute of Psychiatry, Psychology and Neuroscience, King’s College London, London SE5 8AE, UK; 5Department of Experimental Biomedicine and Clinical Neuroscience, University of Palermo, 90129 Palermo, Italy; 6Department of Medical and Surgical Science, Psychiatry Unit, Alma Mater Studiorum Università di Bologna, 40126 Bologna, Italy; 7INSERM U955, Equipe 15, Institut National de la Santé et de la Recherche Médicale, 94010 Créteil, France; 8Department of Child and Adolescent Psychiatry, Hospital General Universitario Gregorio Marañón, School of Medicine, Universidad Complutense, IiSGM (CIBERSAM), 28007 Madrid, Spain; 9Etablissement Public de Santé Maison Blanche, 75020 Paris, France; 10Department of Psychiatry, Early Psychosis Section, Academic Medical Centre, University of Amsterdam, 1105 AZ Amsterdam, The Netherlands; 11Department of Medicine, Psychiatry Area, School of Medicine, Universidad de Oviedo, Centro de Investigación Biomédica en Red de Salud Mental (CIBERSAM), 33006 Oviedo, Spain; 12Barcelona Clinic Schizophrenia Unit, Hospital Clinic of Barcelona, University of Barcelona, IDIBAPS, CIBERSAM, 08036 Barcelona, Spain; 13Department of Psychiatry, School of Medicine, Universidad de Valencia, Centro de Investigación Biomédica en Red de Salud Mental (CIBERSAM), 46010 Valencia, Spain; 14Department of Psychiatry, Servicio de Psiquiatría Hospital “Virgen de la Luz,” 16002 Cuenca, Spain; 15Department of Psychiatry, Psychiatry Genetic Group, Instituto de Investigación Sanitaria de Santiago de Compostela, Complejo Hospitalario Universitario de Santiago de Compostela, 15706 Santiago de Compostela, Spain; 16Division of Psychiatry, Department of Neuroscience and Behaviour, Ribeirão Preto Medical School, University of São Paulo, São Paulo 14049-900, Brazil; 17Department of Preventative Medicine, Faculdade de Medicina FMUSP, University of São Paulo, São Paulo 01246-903, Brazil; 18Rivierduinen Institute for Mental Health Care, 2333 ZZ Leiden, The Netherlands; 19Department of Psychiatry, University of Cambridge, Cambridge CB2 0SZ, UK; 20Psylife Group, Division of Psychiatry, University College London, London W1T 7NF, UK; 21Department of Psychiatry and Neuropsychology, School of Mental Health and Neuroscience, South Limburg Mental Health Research and Teaching Network, Maastricht University Medical Centre, 6200 MD Maastricht, The Netherlands; 22Department of Psychosis Studies, Institute of Psychiatry, King’s College London, London SE5 8AF, UK; 23Department of Health Service and Population Research, Institute of Psychiatry, King’s College London, London SE5 8AF, UK

**Keywords:** childhood adversities, early parental death, early bereavement, ethnic minorities, psychosis, schizophrenia, population-based, case-control, multi-country

## Abstract

Evidence for early parental death as a risk factor for psychosis in offspring is inconclusive. We analyzed data from a six-country, case-control study to examine the associations of early parental death, type of death (maternal, paternal, both), and child’s age at death with psychosis, both overall and by ethnic group. In fully adjusted multivariable mixed-effects logistic regression models, experiencing early parental death was associated with 1.54-fold greater odds of psychosis (95% confidence interval (CI): 1.23, 1.92). Experiencing maternal death had 2.27-fold greater odds (95% CI: 1.18, 4.37), paternal death had 1.14-fold greater odds (95% CI: 0.79, 1.64), and both deaths had 4.42-fold greater odds (95% CI: 2.57, 7.60) of psychosis compared with no early parental death. Experiencing parental death between 11 and 16 years of age had 2.03-fold greater odds of psychosis than experiencing it before five years of age (95% CI: 1.02, 4.04). In stratified analyses, experiencing the death of both parents had 9.22-fold greater odds of psychosis among minority ethnic groups (95% CI: 2.02–28.02) and no elevated odds among the ethnic majority (odds ratio (OR): 0.96; 95% CI: 0.10–8.97), which could be due in part to the higher prevalence of early parental death among minority ethnic groups but should be interpreted cautiously given the wide confidence intervals.

## 1. Introduction

Early parental death, when a child experiences the loss of a parent before age 18, is one of the most severe and distressing stressors a person can experience and a potential risk factor for adult psychopathology. However, current evidence on the relation between early parental death and psychosis is mixed and inconclusive [1,2]. Between 1943 and 1963, eight studies reported a higher prevalence of early parental death among those with psychosis than those without psychosis, and five studies reported the opposite [1]. A meta-analysis of eight studies since 1980 found that early parental death was associated with 1.70-fold greater odds of psychosis, though this estimate was relatively imprecise (95% confidence interval (CI): 0.82–3.52) [2]. A recent population-based case-control study from the United Kingdom found a similar association (odds ratio (OR): 1.80, 95% CI: 0.87–3.73) [3], and six of seven studies using Nordic national register data also found similar associations (ORs 1.2–1.3; incidence rate ratios (IRRs) 1.2–2.0) between early loss and an increased risk of psychosis. Though some of these studies relied on the same registers, each used different definitions of early loss, making comparison difficult [4,5,6,7,8,9,10].

Methodological variation might help explain conflicting findings in prior studies, including the selection of the study sample, the comparison group, and heterogeneity in the operational definitions of early parental death and psychosis [1,11,12,13,14,15,16,17,18,19,20,21]. Study samples have more commonly relied on non-representative clinic patients, e.g., [13,15,16,19,20], and comparison groups have often used clinic patients with other diagnoses or non-randomly recruited controls, e.g., [1,11,13,19,20]. For operational definitions, early loss has ranged from the first two years of life up to age 18, sometimes it has been combined with the death of siblings and/or extended family, and sometimes it has been combined with long-term separation from parents, e.g., [11,15]. For psychosis, many of these studies preceded the publication of standardized diagnostic criteria in the Diagnostic Statistical Manual Third Edition (DSM-III) in 1980 and therefore might not align with current classifications, e.g., [11,15]. This study builds on prior findings by using a recent, ethnically diverse, six-country, population-based sample of incident cases and controls, restricting analysis to early parental death until age 17 and using standardized diagnoses of psychotic disorders.

Further reasons for conflicting findings could be due to additional factors related to early parental death that also influence the association with psychosis such as the type of death (maternal death, paternal death, both deaths) and the child’s age at death. Current evidence is inconclusive for whether there are differences between maternal and paternal death, with some studies finding no differences and some finding that maternal death has a stronger association [1,8,11,13,16,18,19,20,22]. Only two prior studies have considered differences between the death of one or both parents, with one finding no difference and one finding that the death of both parents has a stronger association [8,23]. Current evidence is also inconclusive for differences by child’s age at death, with some studies finding the associations are similar across ages and some finding that children at the earliest ages (before the age of five) are at greatest risk [1,8,11,13,16,18,19,20,22].

It is also important to stratify by ethnic group, since it is a well-established that rates of psychotic disorder are higher in some minority ethnic groups [24,25,26,27]. Given that some minority ethnic groups also have earlier mortality rates [28,29,30], early parental death could be a contributor to this excess risk of psychosis. Only one study from the United Kingdom has explored differences in the association between early parental death and psychosis by ethnic group, finding no differences in the prevalence of early parental death or the magnitude of the association [23]. However, it was likely underpowered, so replication in a larger sample could offer additional insights.

The specific aims of the present study were to (1) determine the association between any early parental death and psychosis; (2) investigate differences of type and timing of early parental death on psychosis; and (3) explore differences in these associations by ethnic group.

## 2. Experimental Section

### 2.1. Study Population

The European Network of National Schizophrenia Networks Studying Gene-Environment Interactions (EU-GEI) collected population-based incidence data on all individuals who came into contact with specialist mental health services with a first-episode psychotic disorder in 16 catchment sites across six countries (Brazil, France, Italy, Netherlands, Spain, and the United Kingdom) from May 2010 to April 2015 [25]. A subset of these incident cases was recruited for a concurrent case-control study, with controls selected from the same catchment areas as the incident cases. In this analysis, we excluded 36 cases from one of the French sites (where no control participants were recruited) and 42 participants who were missing data on early parental death for a final analytic sample of 2549 participants (1072 cases, 1477 controls). Excluded participants had similar characteristics (e.g., age, sex, ethnic group, parent social class, education level, and current employment status) as those included for analysis. Written informed consent was obtained from all participants. Ethical approval was provided by research ethics committees in each site.

### 2.2. Variables

#### 2.2.1. Case/Control Status

Cases included all individuals age 18 to 64 years who made contact with specialist mental health services with first-episode psychosis in the defined catchment areas during the study period (median 25 months, range 12–48 months). This was defined by at least one positive psychotic symptom for at least one day or two negative psychotic symptoms for at least six months. Research diagnoses were made by study staff using the International Classification of Diseases, Tenth Edition (ICD-10) codes F20-F33 (e.g., schizophrenia, schizoaffective disorder, and bipolar disorder). Individuals were excluded if they previously made contact with mental health services for psychosis or if there was any evidence that their psychotic symptoms were due to an organic cause or acute intoxication. Controls were volunteers selected from the same catchment areas using a mixture of random and quota sampling to maximize the representativeness of samples in each catchment area, including randomly selecting from publicly available housing lists and general practitioner lists in some sites and more ad hoc approaches (e.g., internet and newspaper advertisements, leaflets at local stations, shops, and job centers) in others. Some sites also oversampled minority ethnic groups to enable subsequent sub-group analyses; sampling weights were created to account for this in analysis. Controls were excluded if they had been previously diagnosed with or treated for any psychotic disorder [31].

#### 2.2.2. Early Parental Death

Information on early parental death was collected using the Childhood Experiences of Care and Abuse (CECA), a retrospective interview measure [32]. First, questions asked participants to identify their main mother and father figures (including biological parents, step-parents, grandparents, and other). Next, questions enquired whether one or both of the participants’ parents died before age 17, whether it was their mother and/or father, and how old they were at the time of each death. These questions assumed all participants have opposite-sex parents. Consistent with prior studies, three variables were created for analysis: (1) Any parental death (yes/no); (2) type of death (mother, father, both); and (3) child’s age at death (0–5 years, 6–10 years, or 11–16 years) [23].

#### 2.2.3. Ethnic Group

Participants endorsed ethnic categorizations relevant to each country’s context, which were then combined into six categories for standardization across sites: Asian, Black, Mixed, North African, White, and Another. A binary variable was created to distinguish the ethnic majority (White) and minority ethnic groups (Asian, Black, Mixed, North African, Another), as we were interested in assessing whether there was a difference in the association across all minority ethnic groups in comparison to the ethnic majority.

#### 2.2.4. Other Variables

Potential confounders were collected at assessment: Age (continuous), sex (male/female), and parent history of psychosis (yes/no).

### 2.3. Statistical Analysis

All analyses were conducted in Stata 15 [33]. The frequency distributions of sociodemographic characteristics of participants were explored. Continuous variables were expressed as mean ± standard deviation (SD). Categorical variables were expressed as number (percent, %). Pearson’s chi-square and Student’s t tests were used to compare sociodemographic characteristics, any early parental death, type of death, and child’s age at death among cases versus controls and minority ethnic groups versus the ethnic majority.

For Aim 1, to test the hypothesis that there is an association between early parental death and psychosis, multivariable mixed-effects logistic regression models were used after accounting for clustering by catchment site (Stata command *melogit* with catchment site as the level 2 variable) and for the oversampling of minority ethnic groups among the controls relative to the underlying populations at risk by using inverse probability weights (Stata command *pweight*). First, a parsimonious model was run, only adjusting for age and sex. Next, a fully adjusted model was run, taking into account confounding variables selected a priori based on their established relationships with early parental death and psychosis (age, sex, ethnic group, and parental history of psychosis).

For Aim 2, to test the hypotheses that maternal death has a stronger association than paternal death, that the death of two parents has a stronger association than the death of one parent, and that those who experienced parental death at the earliest ages have a stronger association than those who experienced it at older ages, the next set of models separately assessed the association of type of death (maternal, paternal, and both) and child’s age at death (0–5, 6–10, and 11–16) with psychosis. Models for child’s age at death were restricted to those who reported parental death (*n* = 218), using the lowest age tertile (0–5 years) as the referent group given the hypothesis.

For Aim 3, to test the hypothesis that the associations between early parental death and psychosis were stronger among minority ethnic groups than the ethnic majority, we first tested whether the addition of cross-products between ethnic group and early parental death improved the fit of the parsimonious models adjusting for age and sex. Next, we stratified the parsimonious models to present the association separately for the minority ethnic groups and the ethnic majority (graphical outputs created with Stata package *coefplot*).

## 3. Results

In this sample, 42% were cases (i.e., received a diagnosis of first-episode psychotic disorder; 70% non-affective, 28% affective, and 2% unspecified diagnoses). Cases and controls differed on most measured sociodemographic characteristics. Cases were more likely to be younger, men, an ethnic minority, and to have parents who had psychosis (Table 1).

### 3.1. Prevalence of Early Parental Death

Experiencing any early parental death was more common among cases than controls (10.6% vs. 7.0%, *p* < 0.001). Experiencing only maternal death (3.3% vs. 1.7%, *p* = 0.005) or only paternal death (6.2% vs. 4.8%, *p* = 0.005) were both more common among cases than controls. Experiencing both deaths was rare (*n* = 21, 0.8%) but more common among cases than controls (1.2% vs. 0.5%, *p* = 0.005). There was no evidence of differences by child’s age at the time the parent(s) died and odds of psychosis (Table 1).

### 3.2. Early Parental Death and Odds of Psychosis

Table 2 shows the associations between early parental death and psychosis. First, any early parental death before age 17 was associated with 1.54-fold greater odds of psychosis (95% CI: 1.23, 1.92), after adjusting for age, sex, ethnic group, and parent history of psychosis. Second, there were differences by the type and timing of early parental death and psychosis. Individuals who experienced maternal death had 2.27-fold greater odds (95% CI: 1.18, 4.37), and those who experienced paternal death had 1.14-fold greater odds (95% CI: 0.79, 1.64) of psychosis compared with those with no early parental death. Third, individuals who experienced the death of both parents had 4.42-fold greater odds of psychosis (95% CI: 2.57, 7.60) compared with those with no early parental death. Finally, those who were 11–16 years old had 2.03-fold greater odds (95% CI: 1.02, 4.04) and those who were 5–10 years old had 1.26-fold greater odds (95% CI: 0.53, 3.00) of psychosis compared with those who were aged five years or younger at the time of early parental death.

### 3.3. Differences by Ethnic Group

Adding the cross-product terms with ethnic group did not improve model fit for any early parental death (χ^2^ = 0.17, *p* = 0.68), type of death (χ^2^ = 1.65, *p* = 0.65), or child’s age at death (χ^2^ = 0.83, *p* = 0.77). The stratified analyses also indicated that the associations between early parental death and psychosis were broadly similar across minority ethnic groups and the ethnic majority (Figure 1 and Figure 2), with one possible exception. Among minority ethnic groups, those who experienced the death of both parents had 9.22-fold great odds of psychosis (95% CI: 2.02–28.02), while among the ethnic majority, the death of both parents was not associated with elevated odds of psychosis (OR: 0.96; 95% CI: 0.10–8.97). This noted, these findings should be interpreted cautiously given the small cell sizes and wide confidence intervals.

## 4. Discussion

In this multi-country, population-based, incident case-control study, early parental death was associated with increased odds of psychosis. This estimate (OR: 1.54, 95% CI: 1.23, 1.92) was in line with the one derived from meta-analysis (OR: 1.7, 95% CI: 0.82–3.52) [2]. These positive findings corroborate the recent population-based case-control study from the United Kingdom [3] and multiple Nordic register-based studies, although most of the register-based studies included sibling death and had differing age ranges for exposure [4,5,6,7,8,9,10]. The one register-based study that did not find strong evidence of an association (hazard ratio (HR): 1.15, 95% CI: 0.92–1.45) is not comparable here, as it focused on death of the mother’s first-degree relatives and only assessed the child until the age of two [6].

### 4.1. Type and Timing of Early Parental Death

#### 4.1.1. Maternal versus Paternal Death

When considering which parent died, there was around two-fold increased odds of psychosis if it was the mother, but minimal increased odds if it was the father. This aligns with a similar population-based case-control study from the United Kingdom that also found higher odds of psychosis for maternal death and, at most, a modest association for paternal death [23]. However, these findings differ from two Nordic register-based studies that did not find any differences in odds of psychosis between maternal and paternal death [8,9]. Another register-based study mentioned higher odds for maternal than paternal death but did not report the results [7]. Classic psychological theorizing suggests that the mother is the most important attachment figure, e.g., [34]. Further theorizing implicates socialized gender roles, wherein mothers may be able to take on the ‘breadwinner’ role of a lost father while fathers may not be able to fully take on the ‘caretaker’ role of a lost mother, e.g., [19]. Further studies should continue to explore potential differences between maternal and paternal death, noting that these hypothesized pathways will differ across both time and place.

#### 4.1.2. Single versus Both Deaths

Experiencing the death of both parents had around four-fold increased odds for psychosis than experiencing no early parental death. This differs from a similar population-based case-control study from the United Kingdom that did not find that the death of both parents was associated with increased odds of psychosis, although their analysis was likely underpowered since only nine individuals had experienced the death of both parents [23]. However, these findings corroborate the only Nordic register-based study to look at this association, which also found that death of both parents (IRR: 1.79, 95% CI: 1.46–2.19) had a stronger association than single death (IRR: 1.37, 95% CI: 1.30–1.45) on psychosis [8]. Future studies should continue to separately analyze the death of both parents, since the consequences will likely differ from the death of one parent; for example, getting new caretakers versus continuing to live with the surviving parent.

#### 4.1.3. Child’s Age at Parental Death

The current evidence does not suggest any variation in the association between early parental death and psychosis depending on the child’s age at parental death. This aligns with two prior studies that also did not find any differences by age [7,23], although it differs from two other studies that found greatest risk at earliest ages [8,10]. The timing of childhood adversities are postulated to matter given rapid changes in brain development, as well as cognitive and emotional growth [4,22], with the first five years of life and early adolescence both considered potential sensitive periods, e.g., [20]. Future studies should continue to test for differences in timing and include adolescence as a potential sensitive period.

### 4.2. Differences by Ethnic Group

Early parental death was twice as prevalent among minority ethnic groups as the ethnic majority in the controls. This higher prevalence held for the subcategories of maternal death, paternal death, and the death of both parents. In stratified analyses, the magnitude of almost all the associations were broadly similar for both groups. This differs from the only prior study to consider differences by ethnic group, which did not find a higher prevalence of early parental death among minority ethnic groups. However, their findings parallel ours insofar as their associations also had similar magnitudes across ethnic groups [23]. The one striking difference was for the death of both parents. Given that less than one percent of the sample experienced both deaths, these estimates may be unstable, and we interpret them cautiously. However, the death of both parents was almost ten times more common among minority ethnic groups than the ethnic majority in the controls. In stratified analyses, minority ethnic groups who experienced the death of both parents had 9.22-fold greater odds of psychosis (95% CI: 3.03, 28.02), while the ethnic majority did not have elevated odds of psychosis. Because early parental death is rare, there were not sufficient data for subgroup analyses, but future studies should also investigate how this association varies across each of the distinct minority ethnic groups.

### 4.3. Limitations

A major limitation of cross-sectional analyses is the limited inference about causality and potential for reverse causation. In this analysis, temporal ordering was relatively clear because the exposure of early parental death was limited to prior to age 17 and participants had to be 18–64 years when they first received a diagnosis of psychosis (although symptoms could have begun prior to that). However, since participants were recruited based on the outcome, there was still risk of recall bias when retrospectively reporting the exposure. Restricting early parental loss to the narrow and discrete experience of parental death should help minimize this risk [19]. There was also limited power given the rareness of early parental death. Case-control studies are one the most feasible study designs for rare outcomes such as psychosis, but they are not optimal for rare exposures. The measurement of the exposure was missing the cause of death, which is integral because certain causes of early parental death such as suicide, accidents, homicides, and unexpected deaths due to natural causes are associated with greater risk of psychosis than other causes of death [13,19,22]. Finally, there could have been unmeasured confounding, including for parent age at birth and parent social class. We did not adjust for parent social class, since lower parent social class can also follow from early parental death [19,35].

### 4.4. Implications

Despite robust evidence that socioenvironmental factors increase the risk of psychosis [18,36], the evidence on the role of early parental death has been mixed and inconclusive [1,2]. Though early bereavement often leads to intense psychological distress, this resolves over time for most people [22,37]. The singular event of early parental death alone is unlikely to predispose an individual to adult psychopathology—this would rather be the due to additional factors associated with the death [13,19,20,22] and due to an increased exposure to and a vulnerability to other stressors later in life [1,8,11,14,20,22]. This will be influenced by: (1) Consequences: Early parental death leads to a cascade of consequences that are likely to have a more substantive effect on the risk of psychosis than the death itself; and (2) antecedents: Socially disadvantaged groups are at greater risk of early parental death and its consequences, which may help explain inequities in psychosis incidence and outcomes.

#### 4.4.1. Cascading Consequences

Circumstances following early parental death are likely to be better indicators of the effects of early parental death on adult health outcomes than the death itself [13,19,20,22]. These circumstances may buffer or exacerbate conditions by mediating or moderating the effects between early parental death and adult psychopathology. These effects will also depend in part on how circumstances change from before to after the early parental death. Examples of these circumstances include the quality of the parental relationship and parental care prior to death; the emotional response of surviving family members following death; the presence or absence of siblings who provide support or require care; individual abilities and social resources for coping; engagement in health promoting or risking behaviors (including cannabis use); the presence of other childhood adversities; stability or disruption in routines, caretakers, housing, schooling (including ending up in care or an institution); and changes in economic status [8,11,13,14,19,20,21,22,25,37,38]. We briefly note that these consequences also differ by place and over time [1,19,37,39].

Early parental death can still serve as a useful and measureable marker for these cascading consequences [11,18]. However, this framework of cascading consequences justifies using more precise definitions of early parental loss, as these consequences most likely differ following parental death versus long-term parental separation or the death of other family members. It is also justifies assessment by type and timing of deaths. As briefly discussed earlier, differences between maternal and paternal death could be due to changes in subsequent economic support and care, and the stronger association following the death of both parents could be due in part to greater disruption to life and routines.

#### 4.4.2. Investigating Inequities

Early parental death is not randomly distributed in the population. Socially disadvantaged groups are at greater risk of early parental death, which could help explain some inequities in psychosis. Prior studies have established differences by parent social class and parental pychopathology. A lower social class is linked with a lower life expectancy [1,11,35]. Parent psychopathology leads to higher rates of suicides and accidents [13,19,22], and early parental death due to these causes of death has a higher risk of psychosis than for other causes [4,8,10]. Parent psychopathology is also linked to lower life expectancy [40], not only due to suicides and accidents but also due to natural causes of death [41,42,43]. In this regard, genetic heritability could play a role wherein the parent dies earlier as a consequence of psychotic disorder, and the child also has a predisposition to psychotic disorder.

These ideas can be extended to minority ethnic groups, including both more traumatic causes of death and a lower life expectancy due to natural causes of death [29,30]. Our study found that some minority ethnic groups had a higher prevalence of early parental death—especially the death of both parents—compared with the ethnic majority. A recent population-based national study from the United States confirmed that Black Americans experience earlier, more frequent, and more often multiple parental and sibling deaths [28]. It has been proposed that these earlier and more frequent deaths reflect the effects of structural racism, which in turn can also lead to intergenerational transmission of these effects via the cascade of consequences [28,29,30,44]. Further, these consequences will likely be part of a broader proliferation of stressors that lead to a cumulative disadvantage for minority ethnic groups [29,37,45]. Considering the structural antecedents to early parental death will be an important avenue for future research.

## 5. Conclusions

These results add to the emerging evidence that there is an association between early parental death and odds of psychosis in offspring and that this association varies by the type of death. Future studies should consider both the antecedents that lead to higher rates of early parental death among socially disadvantaged groups and the cascading consequences of buffering and exacerbating factors. To do so, studies will need to include larger and more diverse samples. Future studies should also include more comprehensive measures of early parental death, including by type, timing, cause, and consequences of death. Definitions of early parental death should be expanded to consider same-sex parents and blended families that may have more than two main parent figures. New measures should also be developed to ask specific questions about the cascade of consequences following early parental death, including both the subjective, emotional experiences and the objective, material outcomes. In addition to psychosis, studies can be expanded to look at the association of early parental death with other types of psychopathology and to consider whether the effects are specific to psychosis or more broadly generalized. Together, these will help to explicate the association between early parental death and psychosis and to discern which of the potential mediating and moderating factors matter most.

## Figures and Tables

**Figure 1 jcm-08-01081-f001:**
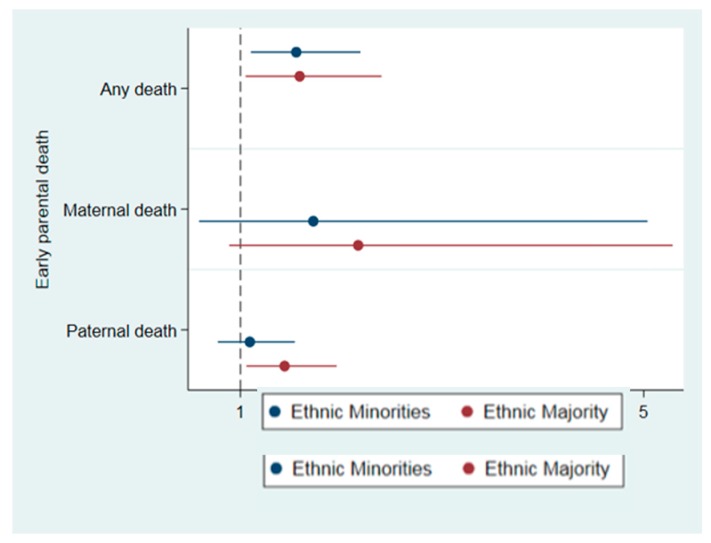
Association between early parental death and odds of psychosis, stratified by ethnic group in the EU-GEI case-control sample. Death of both parents not included due to wide confidence interval with upper-bound value of 28.0.

**Figure 2 jcm-08-01081-f002:**
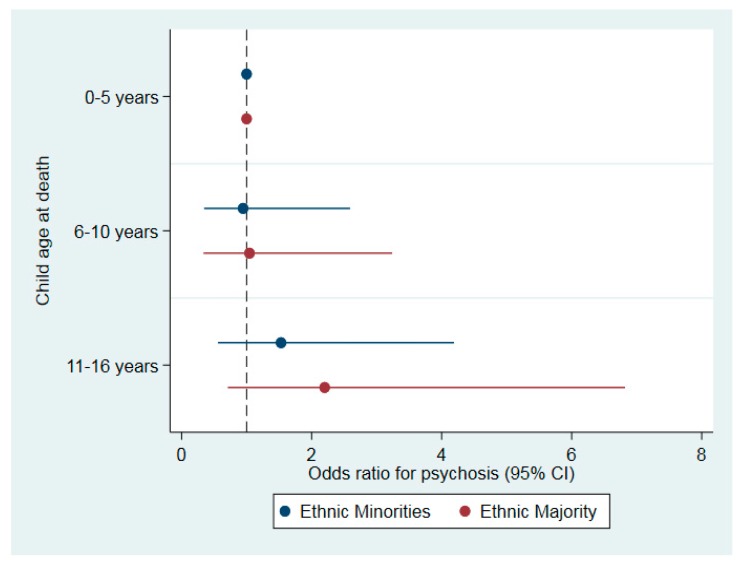
Association between child’s age at early parental death and odds of psychosis, stratified by ethnic group in the EU-GEI case-control sample.

**Table 1 jcm-08-01081-t001:** Sociodemographic characteristics of the European Network of National Schizophrenia Networks Studying Gene-Environment Interactions (EU-GEI) case-control sample by case status and by ethnic group.

	All (*n* = 2549)		Case vs. Control (*n* = 1072 Cases)			Minority vs. Majority (*n* = 698 Minorities)		
	Total *n*	Total %	Case %	Control %	*p*-Value	Min %	Maj %	*p*-Value
Catchment site								
London	416	16.3	17.7	15.3		30.5	11.0	
Cambridge	150	5.9	4.1	7.2		2.3	7.2	
Amsterdam	195	7.7	8.8	6.8		13.3	5.5	
Gouda/Voorhout	209	8.2	9.3	7.4		2.7	10.3	
Madrid	77	3.0	3.6	2.6		1.6	3.6	
Barcelona	67	2.6	2.9	2.4		0.7	3.4	
Oviedo	78	3.1	3.6	2.6		1.9	3.5	
Valencia	80	3.1	4.6	2.1		1.1	3.9	
Créteil	152	6.0	5.0	6.6		11.0	4.1	
Puy de Dôme	61	2.4	1.4	3.1		0.7	3.0	
Bologne	132	5.2	6.3	4.3		2.4	6.2	
Palermo	155	6.1	5.1	6.8		1.6	7.8	
Ribeirão Preto	492	19.3	17.8	20.4		28.5	15.8	
Santiago	66	2.6	2.6	2.6		0.1	3.5	
Verona	163	6.4	5.3	7.2		0.6	8.6	
Cuenca	56	2.2	1.7	2.6		0.9	2.7	
Age at assessment *	34.0 (12.2)		31.3 (10.6)	36.1 (12.9)	<0.001	31.4 (10.9)	35.0 (12.5)	<0.001
Sex								
Female	1193	46.8	38.4	52.9	<0.001	45.7	47.2	0.487
Male	1356	53.2	61.6	47.1		54.3	52.8	
Ethnicity								
White	1850	72.6	64.3	78.7	<0.001	100	0	<0.001
Black	282	11.1	15.3	8.0		0	40.4	
Mixed	223	8.8	10.0	7.9		0	31.9	
Asian	66	2.6	3.1	2.2		0	9.5	
North African	69	2.7	4.2	1.6		0	9.9	
Another	58	2.3	3.2	1.6		0	8.3	
Parental psychosis								
No	2181	96.0	92.8	98.3	<0.001	95.8	96.1	0.741
Yes	90	4.0	7.2	1.7		4.2	3.9	
Parental death								
No	2331	91.5	89.4	93.0	<0.001	86.5	93.4	<0.001
Yes	218	8.5	10.6	7.0		13.5	6.6	
Type of death								
Mother	60	2.4	3.3	1.7	0.005	3.6	1.9	<0.001
Father	158	5.4	6.2	4.8		7.7	4.4	
Both	21	0.8	1.2	0.5		2.2	0.3	
Child’s age								
Age in years *	8.92 (5.11)		9.32 (5.14)	8.49 (5.06)	0.236	9.37 (5.21)	8.52 (5.00)	0.226
0–5 years	62	29.0	26.1	32.0	0.133	26.4	31.2	0.338
6–10 years	58	27.1	23.4	31.1		24.2	29.5	
11–16 years	94	43.9	50.5	36.9		49.4	39.3	

* Mean (SD); for continuous variables, *p*-value was calculated using the student’s *t*-test; for categorical variables, *p*-value was calculated using Pearson’s chi-squared test; due to missing data, *n* may not add to the sample totals.

**Table 2 jcm-08-01081-t002:** Associations between early parental death and psychosis in the EU-GEI case-control sample.

	Unadjusted			Age- and Sex-Adjusted			Fully Adjusted *		
	OR	95% CI	*p*-Value	OR	95% CI	*p*-Value	OR	95% CI	*p*-Value
Parental death									
No	Reference			Reference			Reference		
Yes	1.37	1.09, 1.72	0.007	1.63	1.33, 1.99	<0.001	1.54	1.23, 1.92	<0.001
Type of death									
Mother	1.81	0.89, 3.71	0.103	2.02	1.03, 3.95	0.041	2.27	1.18, 4.37	0.001
Father	1.12	0.89, 1.43	0.330	1.33	1.02, 1.72	0.034	1.14	0.79, 1.64	0.481
Both	2.37	1.14, 4.94	0.022	3.80	2.47, 5.83	<0.001	4.42	2.57, 7.60	<0.001
Child’s age									
0–5 years	Reference			Reference			Reference		
5–10 years	0.85	0.37, 1.98	0.713	0.96	0.45, 2.07	0.924	1.26	0.53, 3.00	0.597
11–16 years	1.68	0.76, 3.72	0.202	1.88	1.03, 3.45	0.040	2.03	1.02, 4.04	0.044

* Separate models were fit for parental death, type of death, and child age at death; each model adjusted for age, sex, ethnic group, and parental history of psychosis.

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
