# Peer review of "Early Parental Death and Risk of Psychosis in Offspring: A Six-Country Case-Control Study"

_jcm, 2019, doi:10.3390/jcm8071081_

Reviewer 1 Report

This is a clearly written manuscript describing an investigation into the risk of parental death in childhood and adolescence for developing a first episode psychosis. Previous studies with smaller or more narrow populations yield differing results and this study aims to clarify the associations. The sample is large and includes participants from 6 countries (5 European and Brazil) with predominantly White majorities but substantial minority populations. The authors use the EU-GEI dataset and also investigate the relative risk of maternal versus paternal versus both parents death in three age groups under age 17. The results indicate that maternal death is a greater risk for development of psychosis and death of both parents places individuals at high risk for developing psychotic illness. This seems to be particularly true for minority individuals.

The authors speculate about possible mechanisms to account for this association between the experience of parental death in childhood and development of first episode psychosis. It is not clear from this study whether they would have been able to include any possible mechanisms . e.g., a cascade effect of risk. It would be interesting to know if there was a direct effect of early stress that is not related to parental psychopathology versus such a cascade of risk  as is the case between early adversity and the development of metabolic syndrome and obesity.

 Author Response

Thank you for your positive comments about our manuscript. We agree that one of the most interesting future questions would be to distinguish between the direct effects of early stress that is not related to parental psychopathology from the cascade of risk that we postulate about in the Discussion section. Unfortunately, we are not able to tease that part with our current data. The EU-GEI study was a cross-sectional design asking adult cases and controls to retrospectively report a broad range of socioenvironmental exposures including from childhood. It does not have specific questions about how participants’ circumstances may have changed from before to after early parental death, and we are not able to determine the temporal order of events based on the measures we do have.

Reviewer 2 Report

This article was well-written, and I appreciate the work put in to this study. As I was reading through, the only comments I would have had were addressed in your Discussion section, particularly when discussing limitations of your study as well as implications. I would like to see research expanding on your findings also including other types of psychopathology.

Author Response

Thank you for your positive comments about our manuscript. We agree that future research should also look at other types of psychopathology, and have added a sentence regarding this into our Conclusions: "In addition to psychosis, studies can be expanded to look at the association of early parental death with other types of psychopathology, and to consider whether the effects are specific to psychosis or more broadly generalized.”